# Hagen-Poiseuille Flow in a Quarter-Elliptic Tube

Mateus D. Bacelar [1], Hugo C. M. G. Ferreira [2], Rajai S. Alassar [3] and André B. Lopes [2,*]

1   Departament of Science and Aerospace Technology, Aeronautics Institute of Technology,
    São José dos Campos 12228-900, SP, Brazil; mateusdutra@outlook.com
2   Departament of Mechanical Engineering, Faculty of Technology, University of Brasília,
    Brasília 70910-900, DF, Brazil; hcmgf1@gmail.com
3   Interdisciplinary Research Center for Renewable Energy and Power Systems, Department of Mathematics,
    King Fahd University of Petroleum & Minerals, Dhahran 31261, Saudi Arabia; alassar@kfupm.edu.sa
*   Correspondence: andre.lopes@unb.br

**Abstract:** We present a rare exact solution of the Navier–Stokes equations for the Hagen–Poiseuille flow through a quarter-elliptic tube. Utilizing the separation of variables method, we derive the solution and report expressions for both the volumetric flow rate and the friction factor–Reynolds number product.

**Keywords:** laminar flow; internal viscous flow; pipe flow; exact solution; Navier–Stokes equations





## 1. Introduction

Fluid flow through a tube with an elliptic geometry is of significant interest in fluid mechanics due to its relevance in various industrial and biological applications, including its widespread use in the heating, ventilation, and air-conditioning industry, particularly in finned-tube heat exchangers [1] and the flow of cerebrospinal fluid in perivascular spaces [2]. The Hagen–Poiseuille flow, a fundamental laminar flow regime, describes the steady flow of an incompressible fluid through a tube under the influence of a constant pressure gradient. The understanding of Hagen–Poiseuille flow in elliptic tubes has evolved over the years through the works of pioneering researchers.

The solution for Hagen–Poiseuille flow in elliptic tubes was first derived in 1868 by Joseph Boussinesq, who found a closed-form analytical expression for the velocity distribution over the elliptic cross-section and for the volumetric flow rate [3]. In 1879, Greenhill resolved the Hagen–Poiseuille flow through confocal elliptic cylinders [4]. After over 13 decades, Alassar and Abushoshah [5] solved the problem of Hagen–Poiseuille flow in wide semielliptic tubes using separation of variables. This work was later complemented by Wang [6], who derived the corresponding expressions for deep semielliptic tubes, and more recently by Kundu and Sakar [7], who analytically and numerically investigated the Hagen–Poiseuille flow through wide and narrow semielliptic annuli. Other notable related works include those of Lal [8] and Chorlton and Lal [9], who explored Couette-Poiseuille flow through a channel with a cross-sectional area bounded by two ellipses of the same ellipticity. Additionally, a recent study by Lopes and Siqueira [10] presented solutions for two related problems: Couette flow and Couette–Poiseuille flow in semielliptic tubes. However, despite significant progress in solving this problem for these various elliptic configurations, the analysis of a pressure-driven flow through a quarter-elliptic tube remains to be explored.

The purpose of this work is to solve the Hagen–Poiseuille flow through a quarter-elliptic tube and to report analytical expressions for the volumetric flow rate and the friction factor–Reynolds number product. This solution represents a rare exact solution of the Navier–Stokes equations. Bazant [11] emphasized the analogies between this phenomenon and sixteen others, spanning six broad fields: fluid mechanics, solid mechanics, heat and mass transfer, stochastic processes, electromagnetism, and electrokinetic phenomena. Thus,

by establishing appropriate correspondences, the presented solution holds the potential to elucidate various physical phenomena, such as the torsion or bending of a beam and the electrostatic potential in a charged cylinder with a quarter-elliptic cross-section.

This work is presented as follows. In Section 2, the governing equation and the boundary condition for this flow are presented and written in dimensionless form. The problem is solved in Section 3 using separation of variables in elliptic coordinates. Qualitative and quantitative results regarding are presented and discussed in Section 4. Section 5 provides a summary of the main contributions of this communication.

## 2. Formulation

Consider the steady Hagen–Poiseuille (pressure-driven) flow of an incompressible Newtonian liquid in a tube of quarter-elliptic cross-section, as depicted in Figure 1. In the sketch, *a* and *b* are the lengths of the semimajor and semiminor axes of the corresponding full elliptic cross-section. The flow is governed by the Poisson equation

$$\nabla^{*2} w^* = -\frac{G^*}{\mu},\tag{1}$$

where $w^* = w^*(x^*, y^*)$ is the fluid velocity in the flow direction, $G^* = -\mathrm{d}p^*/\mathrm{d}z^*$ is the constant pressure gradient, $\mu$ is the dynamic viscosity, and $\nabla^{*2}$ is the Laplacian operator.

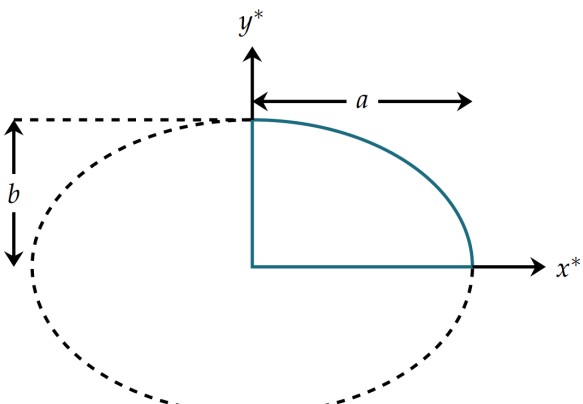

**Figure 1.** Sketch of the quarter-elliptic cross-section.

In the present work, we use $a$ and $a^2 G^*/\mu$ as length and velocity scales, respectively. Therefore, by defining

$$w = \frac{w^*}{\dfrac{a^2 G^*}{\mu}}, \quad x = \frac{x^*}{a}, \quad y = \frac{y^*}{a}, \quad \nabla^* = a\nabla,$$

Equation (1) becomes

$$\nabla^2 w = -1.\tag{2}$$

The aspect ratio,

$$\alpha = \frac{b}{a},\tag{3}$$

is the only resulting dimensionless parameter. Furthermore, due to the no-slip boundary condition, $w = 0$ on any wall.

### 3. Solution

Introducing elliptic coordinates $(\xi, \theta)$ [5,10], Equation (2) takes the form

$$\frac{\cosh^2 \xi_0}{\cosh^2 \xi - \cos^2 \theta} \left( \frac{\partial^2 w}{\partial \xi^2} + \frac{\partial^2 w}{\partial \theta^2} \right) = -1, \tag{4}$$

where $\xi_0 = \tanh^{-1} \alpha$ represents the curved boundary. The boundary conditions are

$$\begin{cases} w = 0 & \text{at} \quad \theta = 0, \\ w = 0 & \text{at} \quad \theta = \dfrac{\pi}{2}, \\ w = 0 & \text{at} \quad \xi = 0, \\ w = 0 & \text{at} \quad \xi = \xi_0. \end{cases} \tag{5}$$

After Alassar and Abushoshah [5], we introduce the following change of variables:

$$w(\xi, \theta) = W(\xi, \theta) - \frac{1}{2} \frac{\sinh^2 \xi \sin^2 \theta}{\cosh^2 \xi_0}. \tag{6}$$

Substituting into Equations (4) and (5), we obtain the Laplace equation:

$$\frac{\partial^2 W}{\partial \xi^2} + \frac{\partial^2 W}{\partial \theta^2} = 0. \tag{7}$$

subject to

$$\begin{cases} W = 0 & \text{at} \quad \theta = 0, \\ W = \dfrac{1}{2} \dfrac{\sinh^2 \xi}{\cosh^2 \xi_0} & \text{at} \quad \theta = \dfrac{\pi}{2}, \\ W = 0 & \text{at} \quad \xi = 0, \\ W = \dfrac{1}{2} \tanh^2 \xi_0 \sin^2 \theta & \text{at} \quad \xi = \xi_0. \end{cases} \tag{8}$$

The problem described by Equations (7) and (8) can be solved by separation of variables. The solution is

$$W(\xi, \theta) = \frac{1}{2\pi} \frac{\sinh \xi_0 \sinh(2\xi) \sin(2\theta)}{\cosh^3 \xi_0} + \sum_{n=1}^{\infty} A_n \sin \left( \frac{n\pi\xi}{\xi_0} \right) \sinh \left( \frac{n\pi\theta}{\xi_0} \right)$$
$$+ \sum_{n=2}^{\infty} B_n \sinh(2n\xi) \sin(2n\theta), \tag{9}$$

where

$$A_n = \frac{(-1)^n \left( 2\xi_0^2 - \pi^2 n^2 \sinh^2 \xi_0 \right) - 2\xi_0^2}{n\pi \left( 4\xi_0^2 + \pi^2 n^2 \right) \cosh^2 \xi_0 \sinh \left( \dfrac{n\pi^2}{2\xi_0} \right)}$$

and

$$B_n = \frac{\tanh^2 \xi_0 \left[ (-1)^n - 2(-1)^n n^2 - 1 \right]}{2\pi(n-1)n(n+1)\sinh(2n\xi_0)}.$$

Thus,

$$w(\xi, \theta) = -\frac{1}{2} \frac{\sinh^2 \xi \sin^2 \theta}{\cosh^2 \xi_0} + \frac{1}{2\pi} \frac{\sinh \xi_0 \sinh(2\xi) \sin(2\theta)}{\cosh^3 \xi_0}$$
$$+ \sum_{n=1}^{\infty} A_n \sin \left( \frac{n\pi\xi}{\xi_0} \right) \sinh \left( \frac{n\pi\theta}{\xi_0} \right) + \sum_{n=2}^{\infty} B_n \sinh(2n\xi) \sin(2n\theta). \tag{10}$$

For the dimensionless volumetric flow rate, we have

$$Q = -\frac{(\pi^2 - 8)}{32\pi} \tanh^3 \zeta_0 + \frac{\zeta_0^2}{\cosh^2 \zeta_0} \sum_{n=1}^{\infty} A_n C_n + \frac{1}{8\cosh^2 \zeta_0} \sum_{n=2}^{\infty} B_n D_n, \tag{11}$$

where

$$C_n = \frac{(-1)^n \left[ \sinh^2 \zeta_0 - \cosh\left(\frac{\pi^2 n}{2\zeta_0}\right) \cosh^2 \zeta_0 \right] + \cosh\left(\frac{\pi^2 n}{2\zeta_0}\right)}{4\zeta_0^2 + \pi^2 n^2}$$

and

$$D_n = \frac{2n(-1)^n - \sinh(2\zeta_0)\sinh(2\zeta_0 n)\left[1 - (-1)^n\right] - 2n\cosh(2\zeta_0 n)\left[(-1)^n \cosh^2 \zeta_0 - \sinh^2 \zeta_0\right]}{(n-1)n(n+1)}.$$

Note that the friction factor–Reynolds number product is related to $Q$ by [12]

$$fRe = \frac{8A^3}{P^2 Q}. \tag{12}$$

Here,

$$A = \frac{\pi \tanh \zeta_0}{4} \tag{13}$$

is the dimensionless cross-sectional area of the flow,

$$P = 1 + \tanh \zeta_0 + E(\mathrm{sech}^2 \zeta_0) \tag{14}$$

is the dimensionless wetted perimeter of the cross-section, and $E(k)$ denotes the complete elliptic integral of the second kind (see, e.g., [13]). It should be noted that $\tanh \zeta_0 = \alpha$ and $\mathrm{sech}^2 \zeta_0 = 1 - \alpha^2$.

In the limiting case of $\alpha \to 1$, it can be shown that

$$Q = \frac{\pi}{24} - \frac{\ln 2}{2\pi}, \tag{15}$$

in agreement with the solution reported by Lopes and Alassar [14], and that

$$fRe = \frac{12\pi^4}{(\pi^2 - 12\ln 2)(\pi + 4)^2}. \tag{16}$$

In the limit of $\alpha \to 0$, it is possible to show that the following relationship holds:

$$fRe = 2\pi^2. \tag{17}$$

It is possible to demonstrate that this limit also holds true for elliptic and semielliptic tubes. This finding is noteworthy as it appears to be a novel contribution to the literature. Shah and London [12] previously reported an approximate value of $fRe$ when $\alpha = 0$ as 19.739 for elliptic tubes and observed that it did not approach the parallel plate geometry. However, they did not acknowledge the possibility of determining its exact value. Furthermore, it is important to note that the results obtained for the velocity field, flow rate, and $fRe$ (as described by Equations (10)–(12)) have been validated through independent methods, namely, finite differences and Rayleigh–Ritz.

Intuitively, one might think that the methodology employed in this section could be extended to any elliptic sector, as previously performed for the elliptic tube [15], for the wide semielliptic tube [5], and for the deep semielliptic tube [6]; however, this is not true. The curves of constant $\zeta$ and $\eta$ in elliptic coordinates are, respectively, confocal ellipses and hyperbolas. In addition, the *x*-axis and *y*-axis are also part of this family of orthogonal

curves. Thus, it is not possible to use the method of separation of variables a priori for any elliptic sector, as it cannot be described using the curves of constant $\xi$ and $\eta$, except in cases where the opening angle of the elliptic sector is $\pi/2$, $\pi$, $3\pi/2$, or $2\pi$. Furthermore, even though elliptic sectors with an opening angle of $3\pi/2$ and $2\pi$ can be formulated in elliptic coordinates (see Appendix A), their boundary conditions are rather exotic, presenting a challenge that makes finding solutions through the method of separation of variables unlikely to work. Thus, it is possible that our work concludes the investigation into tubes whose cross-sections are elliptic sectors that can be analytically solved through the method of separation of variables.

## 4. Results

The dimensionless velocity contours for $\alpha = 0.6$ are shown in Figure 2. The contours span from $0.2u_{\text{max}}$ to $0.8u_{\text{max}}$ in increments of $0.2u_{\text{max}}$, with $u_{\text{max}}$ representing the maximum dimensionless velocity in the sector.

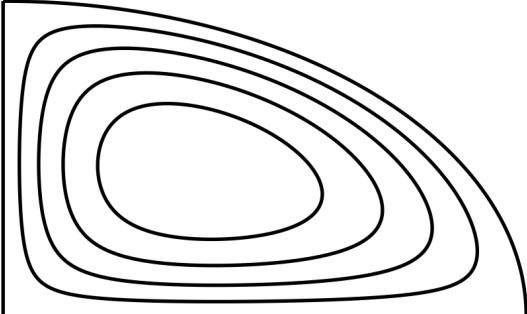

**Figure 2.** Dimensionless velocity contours for $\alpha = 0.6$, showing normalized velocity contour lines ranging from 0.2 to 0.8 with increments of 0.2.

Figure 3 shows the dimensionless volumetric flow rate as of the aspect ratio. The curve is a smooth and gradually increasing function, with flow rate starting at zero and finishing at $\pi/24 - \ln 2/(2\pi) \approx 0.0205$ in the limits of $\alpha \to 0$ and $\alpha \to 1$ (quarter-circular tube), respectively. As the aspect ratio increases, the flow rate rises slowly at first, then more steeply, due to the nonlinear relationship between the two variables. When compared to a circular tube, which has a maximum dimensionless flow rate of $\pi/8 \approx 0.3926$ [3], the flow rate in the quarter-circular tube is much lower, with a ratio of approximately 1:19. This large difference can be attributed not only to the fact that the cross-sectional area of the circular tube is four times larger than that of the quarter-circular tube, but also to the additional retardation in fluid velocity caused by the no-slip condition at the horizontal and vertical walls, as well as the curved wall of the quarter-circular tube.

The relationship between the friction factor–Reynolds number product ($fRe$) and the aspect ratio $\alpha$ for the quarter-elliptic tube is displayed in Figure 4 and can also be found in Table 1. For comparison purposes, the friction factor–Reynolds number product of the semielliptic tube (inferred from Alassar [16] and Abushoshah [5]) and that of the elliptic tube (cf. Shah and London [12]) were added to the plot. The solid curve depicts the friction factor–Reynolds number product for the quarter-elliptic tube, the dashed curve represents it for the semielliptic tube, and the dotted curve corresponds to the elliptic tube. All three curves exhibit a similar trend, with the friction factor–Reynolds number product decreasing as the aspect ratio increases. It is worth noting that they all start at $fRe = 2\pi^2$, which is the limit when the aspect ratio approaches zero. However, when the aspect ratio is greater than zero, the quarter-elliptic tube consistently displays a lower friction factor–Reynolds number product than semielliptic and elliptic tubes for all aspect ratios. Interestingly, when the aspect ratio of the quarter-elliptic tube is approximately equal to 0.42, the friction factor–Reynolds number product is equal to 16, which is the friction factor–Reynolds number product of the circular tube. A parallel behavior is observed in semielliptic tubes when the

aspect ratio approaches around 0.90. It is also interesting to note that Equation (16) provides the lower limit for the friction factor–Reynolds number product in quarter-elliptic tubes.

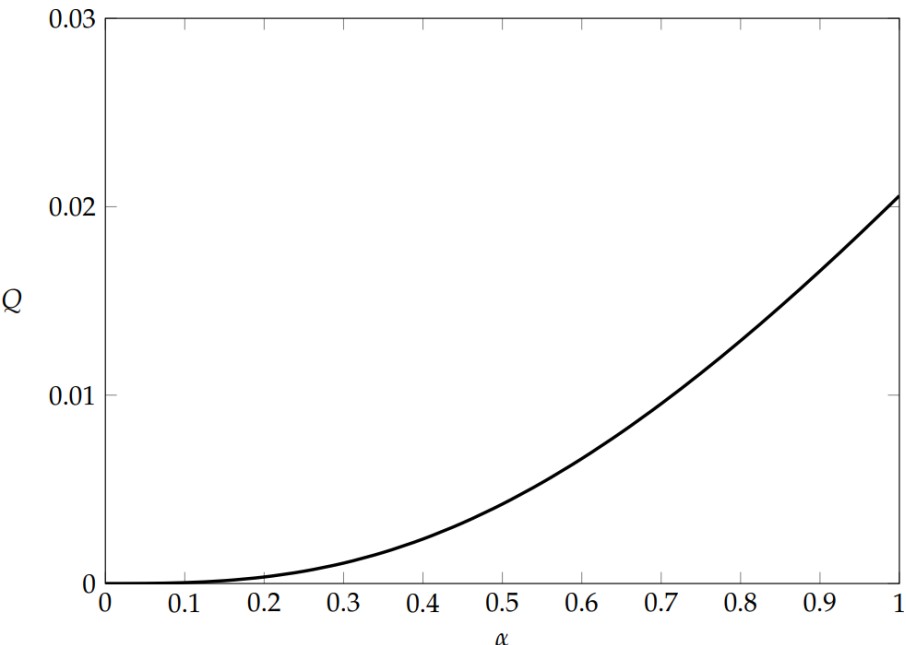

**Figure 3.** Dimensionless volumetric flow rate ($Q$) as a function of the aspect ratio $\alpha$.

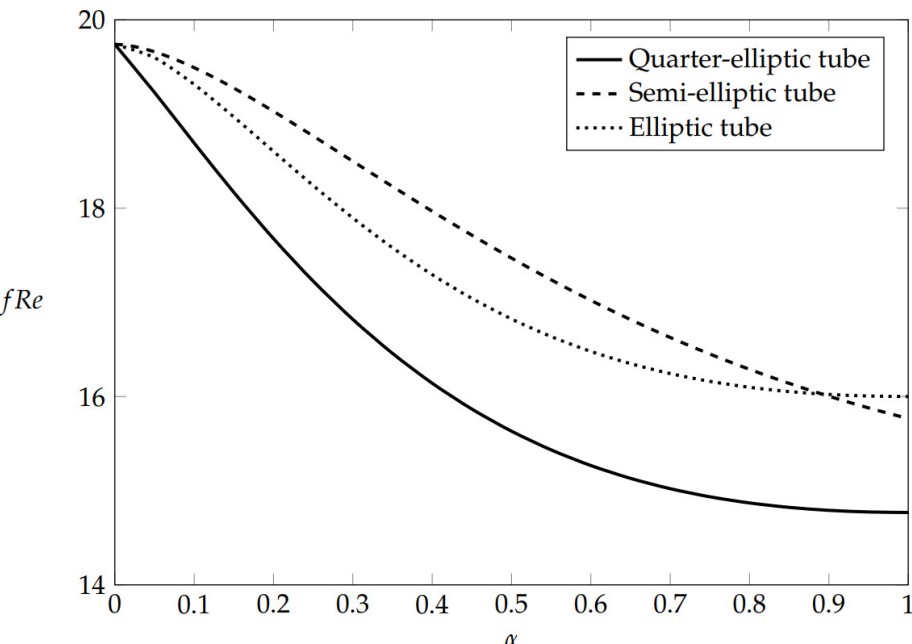

**Figure 4.** Friction factor–Reynolds number product ($fRe$) as a function of the aspect ratio $\alpha$ for the quarter-elliptic tube (solid line), for the semielliptic tube (dashed line), and for the elliptic tube (dotted line).

The $fRe$ factors presented in Table 1 can be approximated (within $\pm 0.05\%$) by

$$fRe = 2\pi^2 \left(1 - 0.5309\alpha - 0.1839\alpha^2 + 1.4313\alpha^3 - 1.4616\alpha^4 + 0.4933\alpha^5\right). \tag{18}$$

We observe that the coefficients of this fifth-degree polynomial equation, which was derived using the least squares method, were chosen to satisfy Equation (17).

**Table 1.** $fRe$ values for quarter-elliptic tubes of aspect ratio $\alpha$.

| $\alpha$ | $fRe$ |
|---|---|
| 0 | 19.7392 * |
| 0.1 | 18.6916 |
| 0.2 | 17.6764 |
| 0.3 | 16.8191 |
| 0.4 | 16.1413 |
| 0.5 | 15.6318 |
| 0.6 | 15.2672 |
| 0.7 | 15.0214 |
| 0.8 | 14.8700 |
| 0.9 | 14.7916 |
| 1 | 14.7687 * |

Closed forms are indicated by asterisks.

## 5. Conclusions

In conclusion, by using the method of separation of variables, we derived analytical expressions that describe the behavior of fluid flow through a quarter-elliptic tube. The volumetric flow rate increases as the aspect ratio increases, but it is much lower than that of a circular tube due to the smaller cross-sectional area and the no-slip condition at the walls. The friction factor–Reynolds number product decreases as the aspect ratio increases, and, except for in the limit where the aspect ratio approaches zero, it is always lower for the quarter-elliptic tube than for the elliptic tube. In this limit, the friction factor–Reynolds number product of both tubes is the same, equal to $2\pi^2$. A fifth-degree polynomial equation was derived to approximate the $fRe$ factors with high accuracy. Overall, the results of this study offer valuable insights into the fluid dynamics of quarter-elliptic tubes and can be useful for the design and optimization of microfluidic devices [17]. Additionally, the study sheds light on the application of wedge-shaped passages, which have been utilized in conventional heat-exchanger applications [18].

**Author Contributions:** Conceptualization, A.B.L. and R.S.A.; methodology, A.B.L. and R.S.A.; software, H.C.M.G.F. and M.D.B.; validation, H.C.M.G.F. and M.D.B.; formal analysis, A.B.L. and R.S.A.; investigation, H.C.M.G.F. and M.D.B.; resources, A.B.L. and R.S.A.; data curation, A.B.L.; writing—original draft preparation, A.B.L.; writing—review and editing, A.B.L., M.D.B. and R.S.A.; visualization, M.D.B.; supervision, A.B.L.; project administration, A.B.L.; funding acquisition, R.S.A. All authors have read and agreed to the published version of the manuscript.

**Funding:** André B. Lopes would like to express his gratitude to the support provided by the Fundação de Apoio à Pesquisa do Distrito Federal (Nº FAPDF: 473/2022, Edital—09/2022 Demanda Induzida).

**Data Availability Statement:** Not applicable.

**Conflicts of Interest:** The authors declare no conflict of interest.

## Appendix A

In this appendix, we formulate the problem of Hagen–Poiseuille flow in two distinct tubes whose cross-section is an elliptic sector. The first tube features a three-quarter elliptic tube (Figure A1a), while the second tube is referred to as a "four-quarter elliptic tube" (Figure A1b) due to a lack of a better term. This particular tube represents the limiting scenario where the opening angle of the elliptical cross-section is equal to $2\pi$, and it is characterized by an infinitely thin wall along the positive $x$-axis. Note that, in all that follows, the variables are presented in a dimensionless form, following the nondimensionalization presented in Section 2.

We begin by noting that Cartesian coordinates $(x, y)$ are related to the elliptic coordinates $(\xi, \theta)$ introduced in Section 3 by

$$x = c \cosh \xi \cos \theta \quad \text{and} \quad y = c \sinh \xi \sin \theta, \tag{A1}$$

where $c = \operatorname{sech} \xi_0$. In the case of the three-quarter-elliptic tube, it follows from Figure A1a and the relations in Equation (A1) that the appropriate boundary conditions for $w$ are

$$\begin{cases} w = 0 & \text{at} \quad \theta = 0, \\ w = 0 & \text{at} \quad \theta = \dfrac{3\pi}{2}, \\ w = 0 & \text{at} \quad \xi = 0 \quad \text{for} \quad 0 < \theta \leq \dfrac{\pi}{2}, \\ w = F & \text{at} \quad \xi = 0 \quad \text{for} \quad \dfrac{\pi}{2} < \theta < \dfrac{3\pi}{2}, \\ w = 0 & \text{at} \quad \xi = \xi_0. \end{cases} \tag{A2}$$

Here, $F = F(\theta)$ is an unknown $2\pi$-periodic function such that $F(\frac{\pi}{2}) = F(\pi) = 0$. As for the case of the cross-section depicted in Figure A1b, the boundary conditions are

$$\begin{cases} w = 0 & \text{at} \quad \theta = 0, \\ w = 0 & \text{at} \quad \theta = 2\pi, \\ w = 0 & \text{at} \quad \xi = 0 \quad \text{for} \quad 0 < \theta \leq \dfrac{\pi}{2}, \\ \dfrac{\partial w}{\partial \xi} = 0 & \text{at} \quad \xi = 0 \quad \text{for} \quad \dfrac{\pi}{2} < \theta < \dfrac{3\pi}{2}, \\ w = 0 & \text{at} \quad \xi = 0 \quad \text{for} \quad \dfrac{3\pi}{2} \leq \theta < 2\pi, \\ w = 0 & \text{at} \quad \xi = \xi_0. \end{cases} \tag{A3}$$

In both problems, a similar feature occurs at $\xi = 0$. More precisely, in the range where $\pi/2 < \theta < 3\pi/2$, the boundary condition is not of Dirichlet type but, rather, periodic in (A2) and Neumann in (A3). This boundary segment corresponds to $-1 < x < 0$ at $y = 0$, a segment in the interior region of the fluid domain in the $x$-$y$ plane, which renders the application of the method of separation of variables more challenging, if not impossible.

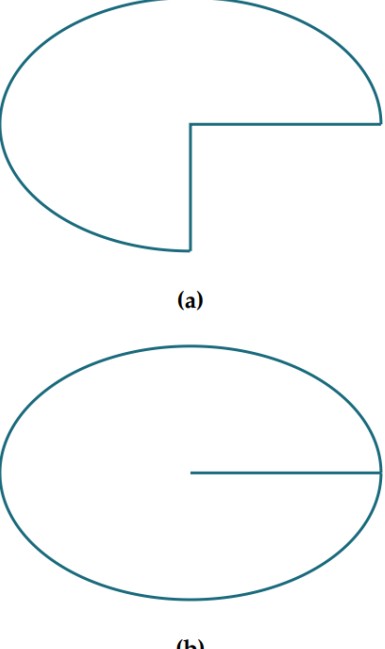

(a)

(b)

**Figure A1.** Sketch of the (**a**) three-quarter-elliptic and (**b**) four-quarter-elliptic cross-sections.

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
