# Peer review of "Hagen-Poiseuille Flow in a Quarter-Elliptic Tube"

_fluids, doi:10.3390/fluids8090247_

Round 1

Reviewer 1 Report

The authors present exact solution of the Navier-Stokes equations for the Hagen-Poiseuille flow through a quarter-elliptic tube. By using elliptic coordinates and separation of variables they derive the solution and present expressions for both the volumetric flow rate and the friction factor-Reynolds number product. The paper is clearly written, the specific elliptic shape can have a nice application to current flows in attics with possible extension into the heat transfer there. So the topic is of interest and the derivations seem correct. The only slight addition I would recommend is possibly expressing the solution back in Cartesian coordinates. The inverse transform of coordinates should not be extremely difficult. Other than that I recommend the paper for publication.

Reviewer 2 Report

The manuscript presents a derivation of the exact solution of the Navier-Stokes equation in the particular case of the quarter-elliptic tube. It is clear in the exposition and presents results on the volumetric flow rate and the friction factor-Reynolds number product.

Regarding the statement at the end of Section 3, I would like to have a clearer explanation as to why the cases presented are the only ones for which the analytical solution could be derived. If the x and y axes are admissible curves, it would feel like the case of the three-quarters ellipse and four-quarters (some kind of ellipse with an infinitely thin wall inside) should also be possible.

At lines 110-112 it is suggested that these results could be useful in the design of microfluidic devices. Is there a possible reference displaying the use of quarter-elliptic tubes?

Reviewer 3 Report

The manuscript considers the exact solution for the Navier-Stokes equations. Finding exact solutions to the equations of hydrodynamics is an urgent problem. The authors constructed an exact solution for a pipe with an elliptical cross section. They presented the material well with formulas and illustrated it well with graphs. I am sure that the manuscript can be published.

Author Response

Thank you for your kind words and positive feedback on our manuscript. Your encouragement further motivates us. We are truly grateful for your assessment and recommendation for publication.

Round 2

Reviewer 2 Report

The authors did a good job addressing my earlier comments. The paper is clearly written, and I think that it can be accepted in its current form except for two minor corrections.

- At line 18 "over 14 decades" is no longer correct due to the newly added reference [3]

- In Equation (A3) there should be xi=0 instead of theta=0 (second to last line)

Author Response

Thank you for your feedback and careful review! We have made the necessary corrections as per your suggestions. The paper has been revised accordingly, addressing the issues you pointed out. We appreciate your time and attention to detail.